# Transcriptomic Responses to Chilling Reveal Potential Chilling Tolerance Mechanisms in Cucumber

**DOI:** 10.3390/ijms232112834

**Published:** 2022-10-25

**Authors:** Xiang Wang, Shuang Mi, Huaiqi Miao

**Affiliations:** Collaborative Innovation Center for Efficient and Green Production of Agriculture in Mountainous Areas of Zhejiang Province, Key Laboratory of Quality and Safety Control for Subtropical Fruit and Vegetable, Ministry of Agriculture and Rural Affairs, College of Horticulture Science, Zhejiang A&F University, Hangzhou 311300, China

**Keywords:** cucumber, chilling response, transcriptome, ethylene, natural variation, CBF

## Abstract

Chilling is a devastating stress that has led to a crisis of production for cucumber (*Cucumis sativus* L.). To determine the molecular mechanisms underlying chilling responses in cucumber, we investigated physiological changes and transcriptomic responses to chilling stress in the chilling-tolerant inbred line CC and chilling-susceptible inbred line R1461. Physiological analysis showed that CC had a higher survival rate, lower H_2_O_2_ accumulation, and ion leakage than R1461 after chilling treatment. RNA-seq analysis identified 938 differentially expressed genes (DEGs) in response to chilling and revealed that chilling stress regulated the transcript levels of genes related to hormones, including auxin, salicylic acid (SA), jasmonic acid (JA), and ethylene. RT-qPCR and pharmacological analysis suggested that cucumber chilling tolerance was associated with variation in the gene expression involved in ethylene biosynthesis and signaling. Exogenously applying 1-aminocyclopropane-1-carboxylic acid (ACC), the precursor of ethylene, improved the chilling tolerance of cucumber, while the exogenous application of the ethylene inhibitor AgNO_3_ impaired the chilling tolerance of cucumber. After ACC treatment, the difference in chilling tolerance between CC and R1461 disappeared, suggesting that the different chilling tolerance level between CC and R1461 is dependent on the ethylene biosynthesis and signaling pathway. In addition, a comparison of cucumber lines with different chilling tolerances revealed that chilling tolerance is highly associated with the up-regulation of *C-repeat binding factor (CBF)* genes, while natural variation in the promoter of *CsCBF1* is associated with chilling response. This study thus provides information on transcriptomic responses in different varieties of chilling-tolerant cucumber and reveals potential chilling tolerance mechanisms that could be used to improve chilling tolerance in cucumber.

## 1. Introduction

Cucumber (*Cucumis sativus* L.), which is one of the most important vegetables, has a tropical origin and is sensitive to chilling temperatures [1,2]. Chilling temperatures often affect cucumber growth and development, limiting its global distribution. Breeding chilling-tolerant cultivars and optimizing cultivation methods are important strategies to avoid chilling injury for cucumber. Dissecting the molecular mechanisms and underlying regulatory network of cucumber responses to chilling stress will provide the necessary theoretical basis for assisting molecular breeding and optimizing cultivation.

Previously studies have indicated that plant hormones and their respective signaling pathways, including auxin (indole-3-acetic acid, IAA) [3], abscisic acid (ABA) [4,5,6], salicylic acid (SA) [7,8], jasmonic acid (JA) [9], and brassinosteroids (BR) [10], play important roles in improving cucumber chilling tolerance. Ethylene is a gaseous plant hormone that also regulates plant chilling stress [11,12,13]. However, this hormone’s role in cucumber chilling tolerance regulation has not been well characterized. Chilling induces ethylene production by inducing the expression of 1-aminocyclopropane-1-carboxylic acid (ACC) synthesis genes in cucumber [10,14,15]. *CsEIN2*, which is a key regulator in the ethylene signaling pathway, was identified as a candidate gene for chilling stress in cucumber [16]. However, whether ethylene has a positive or negative effect on the chilling tolerance of cucumber is little known. In *Arabidopsis thaliana*, ethylene production decreased when plants were exposed to chilling temperature, exogenous ethylene biosynthesis inhibitor aminoethoxyvinyl glycine or loss of function of ethylene signaling genes *ETR1*, *EIN2*, *EIN3*, and *EIN4* enhance *Arabidopsis* freezing tolerance, suggesting that ethylene has a negative effect on freezing tolerance in *Arabidopsis* [13]. In tomato, ethylene production increased upon chilling treatment [12]; exogenous ACC significantly enhanced tomato chilling tolerance; and the *SI-miR164a* functioned upstream of *SINAM3*, which directly activated the expression of *ACS1A*, *ACS1B*, *ACO1*, and *ACO4*, thereby inducing ethylene synthesis upon chilling and thus enhancing tomato chilling tolerance [12]. These results suggested that ethylene plays distinct roles in the regulation of chilling tolerance in different plant species.

*Dehydration-responsive element binding factor 1 (DREB1)/C-repeat binding factor* (*CBF*) genes play a hub role in *Arabidopsis* cold acclimation, where freezing tolerance is enhanced by a prior chilling temperature treatment [17]. In previous studies, *CBFs* were rapidly induced by chilling [18], and their proteins directly bound to the C-repeat/dehydration-responsive (CRT/DRE) cis-elements (CCGAC) of downstream *cold responsive* (*COR*) genes to activate their expressions, thus enhancing the plants’ freezing tolerance [19,20,21,22,23]. *CBF/DREB* genes were also characterized in the regulation of chilling tolerance in cucumber [24,25]. *CsCBF1* and *CsCBF2* transcription was rapidly increased by chilling, while *CsCBF3* was found to be less sensitive in transcriptional regulation to chilling. Overexpression of *CsCBF1-3* in cucumber significantly enhanced the chilling tolerance of cucumber seedlings. In addition, CsCBF1-3 directly activated *CsCOR* gene expression by binding to the cis-elements of their promoters [25]. These results suggest that the function of *CBFs* in regulating chilling tolerance in plants is conserved.

Transcriptomic analysis has been identified as an efficient experimental tool to investigate the genome functions and physiological regulation mechanisms of chilling stress [26,27,28]. Recently, physiological and transcriptome analysis has indicated that the interactions between auxin and hydrogen sulfide signaling are involved in cucumber chilling tolerance [3]. In addition, transcriptome profiling uncovered that exogenous nitric oxide (NO) improves low-temperature tolerance in cucumber by modulating the expression of some key transcription factors and their target genes [27]. These results suggest that transcriptomic analysis plays an important role in dissecting the molecular mechanisms of plant responses to chilling stress.

In this study, we compared the transcriptomics of chilling-tolerant and susceptible inbred lines of cucumber to investigate molecular responses to chilling and the chilling tolerance mechanisms in cucumber. We found that activation of the ethylene signaling pathway and transcriptional regulation of ethylene biosynthesis were critical for chilling tolerance in cucumber. Moreover, the expression variation of *CBF* genes upon chilling was associated with chilling tolerance in cucumber lines, and three polymorphisms of the *CBF1* promoter region conferred an expression variation of *CBF1* upon chilling. Thus, our study expands our understanding of chilling responses and facilitates the genetic improvement of chilling tolerance in cucumber.

## 2. Results

### 2.1. Chilling Tolerance Analysis of Cucumber Inbred Lines

To understand the chilling tolerance of cucumber, we assessed the survival rates of 10 cucumber inbred lines after chilling treatment. Thirty seedlings for each line at the three-leaf stage were exposed to chilling treatment in a 6 °C chamber for 4 days and then transferred to normal conditions for recovery. After 7 days of recovery, CC had the highest survival rate in these inbred lines, whereas R1461 had a medium-to-low survival rate (Appendix A).

We then chose CC and R1461 to investigate the cucumber physiological responses to chilling temperature. After 4 days of chilling treatment (6 °C) and 7 days of recovery, CC displayed a slighter leaf curl and higher survival rate than R1461 (Figure 1A,B). Furthermore, when seedlings of CC and R1461 were exposed to 6 °C for 48 h, CC displayed lower H_2_O_2_ accumulation (Figure 1C) and significantly lower ion leakage than R1461 (Figure 1D). These results strongly suggest that CC is a chilling-tolerant inbred line, while R1461 is a chilling-susceptible inbred line.

### 2.2. Transcriptomic Analysis of CC and R1461 to Chilling Treatment

To determine the molecular responses to chilling and obtain a better understanding of cucumber chilling tolerance, we performed transcriptome profiling on CC and R1461 seedlings under normal conditions and chilling treatment for 3 h. The total reads from R1461 (between 46,000,000 and 50,000,000 per sample) and CC (47,000,000 and 51,000,000 per sample) were obtained by RNA-seq (Appendix A). About 95% of the reads from each sample were uniquely mapped to the cucumber (Chinese Long) v2 Genome (Appendix A). Principal component analysis (PCA) analysis showed that CC and R1461 were similar to each other after both 0 and 3 h chilling treatment in the first two principal components, and chilling treatment was the most important factor in differentiating between samples (Appendix A). In addition, the three biological repeats were always clustered together among the samples except for the spreading of the samples of CC after 3 h chilling treatment (Appendix A). These PCA results indicate the high reproducibility of this dataset. Additionally, the chilling treatment induced a similar global transcriptional change in CC and R1461.

### 2.3. Transcriptomic Response of CC and R1461 to Chilling Temperature

To investigate the responses to chilling at the molecular level, we analyzed the differentially expressed genes (DEGs) and their enriched GO terms for the biological processes (BPs) in CC and R1461 under chilling treatment. More up- and down-regulated DEGs under chilling versus non-chilling (normal) treatment were detected in R1461 than in CC (Figure 2A,B). In total, the Venn diagram analysis revealed that 99 chilling-induced DEGs (74 up- and 25 down-regulated) were unique to CC, while 653 (332 up- and 331 down-regulated) DEGs were unique to R1461 (Figure 2A,B), indicating the specificity of the CC and R1461 to chilling responses and suggesting that these DEGs may be related to the different chilling tolerance levels in CC and R1461. A total of 59 BP terms (51 for up-regulated DEGs, 8 for down-regulated DEGs) were enriched for DEGs in CC or R1461 (Figure 2C,D). CC and R1461 exhibited a differential enrichment of BP terms; 11 BP terms (9 for up-regulated DEGs, 2 for down-regulated DEGs) were unique to CC, while 54 BP terms (50 for up-regulated DEGs, 4 for down-regulated DEGs) were unique to R1461. For up-regulated DEGs, the most significant BP terms for CC and R1461 were “response to chitin” and “cell division”, respectively. For down-regulated DEGs, the most significant BP terms for CC and R1461 were “regulation of growth” and “auxin-activated signaling pathway”, respectively.

It is well known that hormones play critical roles in chilling tolerance in plants [3,5,6,7,8,12,29]. In our data, 7 of the 59 BP terms were related to hormones and their signaling pathways, including “response to salicylic acid”, “response to jasmonic acid”, “response to ethylene”, “induced systemic resistance”, “jasmonic acid mediated signaling pathway”, “ethylene-activated signaling pathway”, and “auxin-activated signaling pathway” (Figure 2C,D), suggesting a large change in hormone responses upon chilling in cucumber. For up-regulated DEGs, the BP term “ethylene-activated signaling pathway” was enriched in both CC and R1461 (Figure 2C). Interestingly, the BP term for up-regulated DEGs “response to ethylene” was uniquely enriched in CC (Figure 2C), and 11 genes belonging to the “ethylene-activated signaling pathway” term were down-regulated in R1461 (Figure 2D). These results suggest that ethylene plays an important role in cucumber chilling responses and that the distinct gene regulation of chilling response in these two BP terms may be the one reason for the different chilling tolerance levels observed in CC and R1461. In addition to ethylene, the BP terms (for up-regulated DEGs) “response to salicylic acid”, “induced systemic resistance”, and jasmonic acid mediated signaling pathway” were enriched in CC and R1461. The BP term (for up-regulated DEGs) “response to jasmonic acid” was uniquely enriched in R1461. The BP term (for down-regulated DEGs) “auxin-activated signaling pathway” was enriched in CC and R1461.

### 2.4. Expression Analysis of DEGs Involved in the Ethylene Biosynthesis and Signaling Pathway

As ethylene signaling is involved in the regulation of plant responses to several stresses, including chilling stress [12,30,31,32,33]. Variation in gene expression in response to chilling in nature may confer chilling tolerance in plants [34]. CC and R1461 featured uniquely enriched terms in ethylene signaling, suggesting that the expression variation of genes involved in ethylene signaling may confer the observed variation in the chilling tolerance of CC and R1461. Therefore, to further investigate the role of ethylene biosynthesis and signaling in response to chilling, we compared the expression levels of DEGs under the CC-unique BP term “response to ethylene” and the R1461-unique BP term “ethylene-activated signaling pathway”. The BP term “response to ethylene” included five genes, all of which were significantly up-regulated in CC. Csa2G355030, encoding an MYB transcription factor, was significantly up-regulated in CC under chilling (Figure 3A). In contrast, Csa2G355030 presented no altered expression in R1461. Although the other genes were also up-regulated in R1461, the fold changes in three of these four genes were less than those in CC (Figure 3A), suggesting that the variation of chilling-induced expression among these genes may confer cucumber chilling tolerance in CC. However, the expression of these four genes was also up-regulated in R1461, and thus the influence of these four genes on chilling tolerance may be less than Csa2G355030. The second examined BP term was “ethylene-activated signaling pathway” (Figure 3B). Four of the genes herein were down-regulated in both CC and R1461. However, the fold change of the down-regulation among these two genes in CC was less than that in R1461. In addition, seven genes in this BP term presented no alteration in CC but were significantly down-regulated in R1461, suggesting that these genes may confer chilling susceptibility in R1461.

We further performed RT-qPCR to assess the expression levels of genes involved in ethylene synthesis under chilling treatment in CC and R1461. The transcript levels of *ACS2* (Csa4G049610) and *ACO2* (Csa6G511860) were significantly increased after exposing cucumber plants to chilling temperatures (Figure 3C,D). However, the chilling-induced expression level changes of both *ACS2* and *ACO2* in CC were larger than those in R1461, suggesting that the variation in ethylene synthesis may confer cucumber chilling tolerance. The CC and R1461 plants were treated with 100 µM exogenous ACC, water, and AgNO_3_ and then exposed to chilling at 6 °C for 4 days to further investigate the role of ethylene in chilling tolerance. As expected, after 7 days of recovery, 100 µM ACC significantly increased the chilling tolerance of CC and R1461 compared to the water treatment, which was used as a control, while 100 µM AgNO_3_ decreased the chilling tolerance of CC and R1461 (Figure 3E). Intriguingly, after 100 µM ACC treatment, R1461 showed similar chilling tolerance to CC. These findings suggest that the different chilling tolerance levels between CC and R1461 are dependent on ethylene level.

### 2.5. Expression Patterns of CsCBFs in CC and R1461 under Chilling Treatments

Studies have shown that *DREB1*/*CBFs* regulate cold acclimation in *Arabidopsis* [17] and chilling tolerance in rice [35]. Overexpression of *CBF1-3* in cucumber was found to significantly increase cucumber chilling tolerance [25]. To gain insights into the chilling responses of *CBF* genes in cucumber, we performed a phylogenetic tree analysis using *DREB1B* in *Arabidopsis* as a query. As shown in Figure 4A, Csa_3G180260 (*CsCBF1*), Csa_5G174570 (*CsCBF2*), and Csa_5G55570 (*CsCBF3*) were close to *DREB1A-F* in *Arabidopsis*. Csa_3G751440 (*CsCBF4*) was close to the *DREB1F* in rice and Solyc01g009440 in tomato. Our RNA-seq data showed that the extent of the up-regulation of *CsCBF1* differed between CC (416 folds, Appendix A) and R1461 (108 folds, Appendix A). *CsCBF2* was significantly up-regulated (22 folds, Appendix A) in CC but showed no altered expression in R1461, suggesting that different extents of up-regulation of the *CBF* genes contributed to distinct chilling tolerance levels in CC and R1461. Further RT-qPCR analysis revealed that all four *CBF* genes were significantly up-regulated after chilling in CC (Figure 4B–E). *CsCBF1* and *CsCBF2* rapidly increased and then decreased after reaching their maximum values at 3 h (Figure 4B,C). However, the expression level of *CsCBF3* showed a continual increase over 24 h (Figure 4D). The expression level of *CsCBF4* showed a lesser change than the others (Figure 4E), suggesting that this gene was less sensitive to chilling temperatures. In R1461, only the expression level of *CsCBF2* and *CsCBF3* changed drastically after chilling (Figure 4C,D). These results suggest that the up-regulation of *CsCBF* genes confers chilling tolerance and that the different expression patterns of these genes in CC and R1461 contribute to their distinct chilling tolerance levels.

### 2.6. Natural Variation in CBF1 Promoter Confers a Distinct Chilling Response

Given the expression variation of *CsCBF1-3* in CC and R1461 (Figure 4B–D) and that the overexpression of the *CsCBF1-3* enhanced cucumber chilling tolerance [25], we then determined whether there are polymorphisms present in their genomic sequences. We sequenced these genes (about 2.8 kb), including the promoter (about 2 kb) and coding regions (about 0.8 kb), via PCR amplification followed by Sanger sequencing. Compared to the Cucumber (Chinese Long) v2 Genome, no differences in *CsCBF2* and *CsCBF3* were found between the CC and R1461 genomic sequences, indicating that the different expression patterns under chilling in CC and R1461 could be caused by the other regulators, which were perhaps located in other quantitative trait loci (QTLs). It also indicates the post-translational regulation of CsCBF2 and CsCBF3. For example, stability and activity differences of both transcription factors. Sequence analysis of *CsCBF1* identified two single nucleotide polymorphisms (SNPs) and one base deletion in the promoter region between CC and the other lines (Figure 5A, Appendix A). Hence, we postulated that these three polymorphisms may contribute to the different expression patterns of *CsCBF1* in CC and R1461 under chilling. To confirm our prediction, the 1.5 kb promoters upstream of the transcriptional start site for *CsCBF1* from CC and R1461 were cloned, and each was fused with the reporter gene *Firefly* luciferase (*LUC*). Another reporter gene, *Renilla* luciferase (*REN*), residing in the same vector, was expressed under control of the CaMV 35s promoter to normalize transformation and expression efficiencies (Figure 5B). The reporter constructs *pCBF1^CC^::LUC* and *pCBF1^R1461^::LUC* were each transformed into protoplasts from the *Arabidopsis* Col-0 plants. As shown in Figure 5C, there was no difference in luciferase activity at normal temperature. Under chilling temperature, both *pCBF1^CC^::LUC* and *pCBF1^R1461^::LUC* presented significantly higher luciferase activity compared to that at normal temperature. In addition, at chilling temperatures, *pCBF1^CC^::LUC* showed significantly higher LUC activity than *pCBF1^R1461^::LUC*. In addition, we examined the *CsCBF1* transcript levels in the nine chilling-susceptible cucumber lines at 0 and 3 h after chilling treatment compared to CC. As expected, the *CsCBF1* transcript level was strongly induced by chilling in CC (Figure 5D), suggesting that natural variations in the promoter region of *CsCBF1* contribute to *CsCBF1* expression variation in response to chilling temperatures. However, there were no differences in *CsCBF1* transcript levels between the CC and chilling-susceptible lines at normal temperatures, suggesting that the influence on gene expression by the polymorphisms in *CsCBF1* was specific to chilling temperatures.

## 3. Discussion

Chilling stress severely affects cucumber plant growth and production [36,37,38]. How cucumber plants respond and adapt to chilling temperature is an important biological question, but the corresponding molecular mechanisms are still poorly understood. Here, we identified an extreme chilling-tolerant cucumber inbred line CC with a high survival rate, low H_2_O_2_ accumulation, and ion leakage after chilling treatment (Figure 1). Transcriptome analysis in a chilling-tolerant inbred line (CC) and a susceptible inbred line (R1462) revealed a variation in gene expression that could contribute to differences in chilling tolerance in cucumber.

Transcriptome analysis identified 285 and 839 DEGs in chilling-tolerant (CC) and chilling-susceptible (R1461) cucumber cultivars, respectively (Appendix A), suggesting that these genes may confer chilling tolerance in cucumber. GO analysis was used to analyze the functions of plant genes in response to low-temperature stress [5,27,28,35,39]. Previous studies found that chilling induces ethylene production in cucumber [14,15]. However, it was not tested in these studies whether ethylene signaling plays a role in cucumber chilling tolerance. In the present study, we found that the BP term “response to ethylene” for up-regulated DEGs was enriched in the chilling-tolerant line CC (Figure 2C). Additionally, we observed that “ethylene-activated signaling pathway” for down-regulated DEGs was enriched in the chilling-susceptible R1461 (Figure 2D), indicating that these two processes may confer distinct chilling tolerance in CC and R1461. In addition, the expression levels of ethylene biosynthesis genes in CC were significantly higher than those in R1461 (Figure 3C,D), suggesting that chilling-induced ethylene production is essential for cucumber chilling tolerance. This result is supported by the similar chilling tolerance presented by CC and R1461 when sprayed with ACC (Figure 3E).

Ethylene signaling is known to be very important for low-temperature stress in *Arabidopsis* and tomato [12,13]. In *Arabidopsis thaliana*, ethylene was found to rapidly decrease under chilling temperatures and negatively regulate plant responses to freezing stress [13]. Conversely, chilling temperatures rapidly induced the expression of ethylene synthesis genes in tomato and increased ethylene production, thereby enhancing tomato chilling tolerance [12]. These results suggest that ethylene plays distinct roles in the regulation of chilling tolerance among different plant species. Hence, there is a need to investigate the roles of ethylene in chilling tolerance regulation in different plant species. In our work, we identified a chilling-responsiveness of transcripts of the two BP terms “response to ethylene” and “ethylene-activated signaling pathway” and revealed the variation in gene expression among these genes or ethylene synthesis genes in chilling-tolerant and chilling-susceptible cucumber inbred lines, suggesting that there are natural variations in ethylene signaling and biosynthesis between these two inbred lines possibly contributing to chilling tolerance. In the future, it would be interesting to generate populations from these two lines to map the QTLs related to chilling tolerance in cucumber. Moreover, we found that spraying with 100 µM ACC can significantly enhance cucumber chilling tolerance (Figure 3E). This treatment could be directly used for cucumber cultivation to avoid damage from chilling temperatures.

*CBF* genes are considered to be a hub in cold acclimation [17]; their roles in cucumber chilling tolerance were also previously characterized [25]. Our study indicated that *CsCBF1-3* genes were more strongly up-regulated by chilling in the cucumber chilling-tolerant line CC than in the chilling-susceptible R1461 (Figure 4B–D), suggesting that the expression variation in *CsCBF1-3* genes upon chilling confers cucumber chilling tolerance. This result is supported by the observation that the overexpression of *CsCBF1-3* significantly enhances cucumber chilling tolerance [25]. On the genetic basis of the two SNPs and one deletion in *CsCBF1* identified in this study, we suggest that *CsCBF1* has great potential for improving cucumber chilling tolerance via molecular breeding techniques. Ethylene-insensitive 3 (EIN3), a nuclear transcription factor, functions downstream in ethylene signaling [40,41]. EIN3 directly binds to *CBF1-3* promoter regions to negatively regulate the expression of *CBFs* and freezing tolerance in *Arabidopsis* [13]. Our work revealed that ethylene plays a critical role in the distinct chilling tolerance levels of CC and R1461. We speculate that other natural variations in *CsCBF* regulators involved in ethylene signaling contribute to cucumber chilling tolerance. Therefore, mapping the genes and dissecting the molecular mechanisms underlying how ethylene regulates *CsCBF* gene expression under chilling stress in cucumber will be an interesting future research direction.

## 4. Materials and Methods

### 4.1. Plant Materials and Chilling Treatment

All the plants used in this study were grown in a mixture of soil and vermiculite (1:1) at (22 ± 2 °C) under long-day conditions (16 h light/8 h dark, 130 μmol/m^2^/s) for 16 days before being sent to the growth chamber (6 °C, 16 h light/8 h dark, 130 μmol/m^2^/s) for chilling treatment. Survival was defined as the plant showing the emergence of new leaves. Survival rate was calculated as surviving seedlings versus the total number of plants. To determine the cucumber seedling survival rate after chilling, 30 seedlings for each replicate at the three-leaf stage were subjected to chilling treatment for 4 days and then moved to normal conditions for 7 days before plant survival was assessed. To determine the cucumber seedling survival rate after chilling with ACC and AgNO_3_ treatment, 30 seedlings for each replicate were grown for 16 days in a solution containing 1/2 MS until the three-leaf stage. Before chilling treatment, plants were sprayed with 100 µM ACC, water, and 100 µM AgNO_3_. After 4 days of chilling treatment, seedlings were moved to normal conditions for 7 days before survival rate was assessed.

### 4.2. DAB Staining

The accumulation of H_2_O_2_ in cucumber seedlings after chilling treatment was detected via DAB staining as described previously, with some modifications [42]. In brief, tissues were collected in 15 mL tubes and incubated in a 1 mg/mL diaminobenzidine (DAB) solution (Sigma, NJ, USA) for 8 h in the dark to determine the accumulation of H_2_O_2_.

### 4.3. Assay of Relative Electrolyte Leakage

Relative ion leakage was previously used as an indicator of membrane damage under chilling [39]. Relative ion leakage in this study was measured as previously described [43], with few modifications. Briefly, 0.1 g leaves were cut into 0.5 cm pieces and incubated in 0.4 M mannitol at 25 °C with gentle shaking for 3 h before the initial conductivity of the solution was measured with a conductivity meter (DDS-307A, Ningbo Hinotek Technology, Ningbo, China). Total conductivity of the solution was measured after incubation at 85 °C for 20 min. An assay was done on 3 biological replicates consisting of 3–6 individual seedlings each, and the ion leakage rate was calculated as the initial conductivity versus total conductivity.

### 4.4. RNA Sequencing and RT-qPCR

The three-leaf-stage (16 days old) seedlings of CC and R1461 were treated at 6 °C for 0 and 3 h, and leaves were sampled with 3 biological replicates consisting of 3–6 individual seedlings each. Total RNA was extracted using TRIzol reagent. RNA integrity was evaluated using an Agilent 2100 Bioanalyzer (Agilent Technologies, Santa Clara, CA, USA). The samples with an RNA Integrity Number (RIN) ≥ 7 were subjected to subsequent analysis. The libraries were constructed using a TruSeq Stranded mRNA LTSample Prep Kit (Illumina, San Diego, CA, USA). Then, these libraries were sequenced on an Illumina sequencing platform (HiSeqTM 2500 or Illumina HiSeq X Ten), and 125 bp/150 bp paired-end reads were generated. The transcriptome sequencing and analysis were conducted by OE Biotech Co., Ltd. (Shanghai, China).

For RT-qPCR, the three-leaf-stage seedlings of CC and R1461 were treated at 6 °C for 0, 3, 6, 12, and 24 h. Leaves were then sampled with 3 biological replicates consisting of 3–6 individual seedlings each. Total RNA was isolated from seedling tissues using TRIzol reagent (Invitrogen, Carlsbad, CA, USA) and treated with DNase (TRANSGEN BIOTECH, http://www.transgen.com.cn) (accessed on 5 March 2022), before being used for cDNA synthesis. Reverse transcription was done using a PrimerScript II 1st Strand cDNA Synthesis Kit (TaKaRa). First-strand cDNA synthesis of the cDNA strand of mRNA was performed using the previously reported method [34]. *ACTIN* was used as the control for mRNAs. RT-qPCR was performed using diluted cDNA on a Step One Plus (ABI) real-time thermocycler. Three independent biological replicates were performed. Gene-specific primers are shown in Appendix A.

### 4.5. RNA-Seq Data Analysis

The clean data were mapped to the Cucumber (Chinese Long) v2 Genome (http://plants.ensembl.org/index.html) (accessed on 7 November 2021), using HISAT2 [44]. Differential expression analysis was performed using the DESeq (2012) R package. A value of *p* < 0.05 and fold change >2 or fold change <0.5 were set as the threshold for significantly different expressions. GO enrichment analyses of DEGs were performed using R base with hypergeometric distribution. The GO terms with q < 0.05 were considered significantly enriched. Venn diagrams and principal component analysis (PCA) were performed on the platform OuyiCloud (https://cloud.oebiotech.cn/task/category/pipeline/) (accessed on 11 November 2021). Enrichment dot bubble plots and heat maps were generated by http://www.bioinformatics.com.cn (accessed on 11 November 2021).

### 4.6. Promoter Activity Assay

The 1.5 kb promoter fragments upstream of *CsCBF1* (Csa3G180260) transcription start site were amplified from CC and R1461 and then cloned into the pGREEN II 0800-LUC reporter vector [45] to generate *pCBF1^CC^::LUC* and *CBF1^R1461^::LUC* plasmids, respectively. Protoplasts were isolated from 14-day-old seedlings of *Arabidopsis* Col-0 plants grown on 1/2 Murashige and Skoog media under long-day conditions. Plasmid DNA was transformed into protoplasts following methods described previously [46]. Transfected protoplasts were incubated at 22 °C for 8 h under dark and then shifted to 6 and 22 °C for 6 h incubation. Firefly and Renilla luciferase activities were measured using a Dual-Luciferase Reporter Assay System (Promega, https://www.promega.com) (accessed on 20 May 2022).

### 4.7. Phylogenetic Tree

The phylogenetic tree of *DREB1* was generated using PhyloGenes (http://www.phylogenes.org) (accessed on 21 Match 2022) with the Arabidopsis *DREB1B* gene as a query on genes from Arabidopsis, rice, cucumber, and tomato.

## 5. Conclusions

In this study, transcriptome analysis in the chilling-tolerant inbred line CC and chilling-susceptible inbred line R1461 revealed that chilling tolerance in cucumber is associated with the ethylene biosynthesis and signaling pathway, as well as the expression variations of *CsCBF1-3* genes under chilling. The expression variation of *CsCBF1* in CC and R1461 upon chilling was associated with natural variations in its promoter region. This study provides more comprehensive information on cucumber chilling response and underscores the potential significance of *CsCBF1* for future molecular breeding chilling tolerance in cucumber.

## Figures and Tables

**Figure 1 ijms-23-12834-f001:**
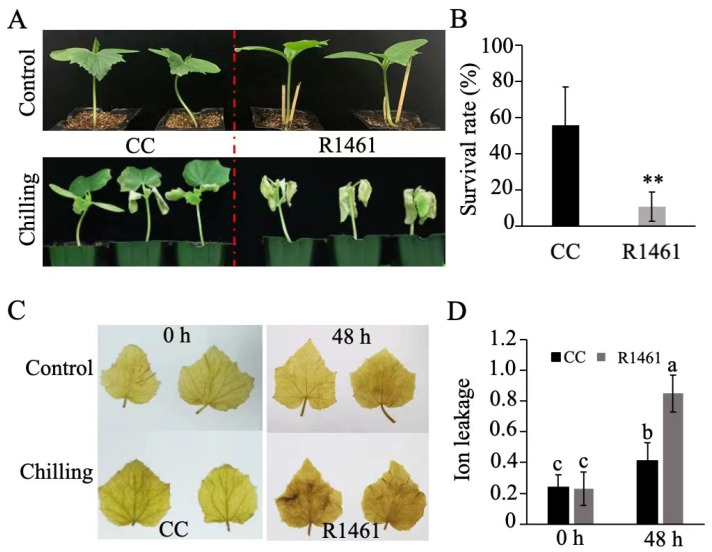
Effects of chilling treatment on CC and R1461. (**A**) Morphological phenotypes of CC and R1461 inbred lines before and after 4 days of chilling treatment. (**B**) Survival rate (%) of CC and R1461 inbred lines after 4 days of chilling treatment. Data are the means ± SD of three biological replicates consisting of 30 seedlings each. ** means *p* < 0.01, Student’s *t*-test. (**C**) DAB staining. (**D**) ion leakage before and after 48 h chilling treatment. Data are the means ± SD of three biological replicates consisting of 30 seedlings each. Different letters above the bars indicate significant differences among samples as determined by a one-way ANOVA/Duncan (*p* < 0.05).

**Figure 2 ijms-23-12834-f002:**
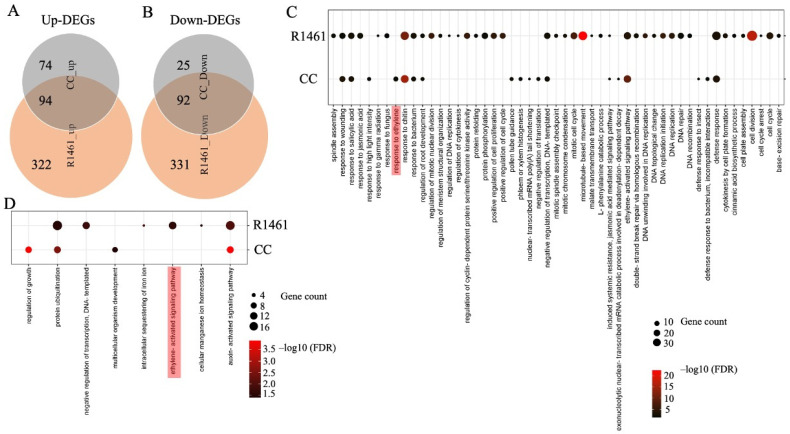
Analysis of chilling-induced differentially expressed genes (DEGs) and their enriched Biological Process (BP) GO terms in CC and R1461. (**A**) Venn diagram analysis of the up-regulated DEGs and down-regulated DEGs (**B**) in CC and R1461 under chilling treatment. The DEGs were identified from chilling versus non-chilling with |log_2_FC| ≥ 1, FDR ≤ 0.05. A dot bubble diagram is presented for enriched BP terms in up-regulated DEGs (**C**) and down-regulated DEGs (**D**) in CC and R1461 under chilling treatment. Significant enrichment of BP terms was established with FDR ≤ 0.05. The *x*-axis represents the enriched BP terms, and the *y*-axis indicates the CC and R1461.

**Figure 3 ijms-23-12834-f003:**
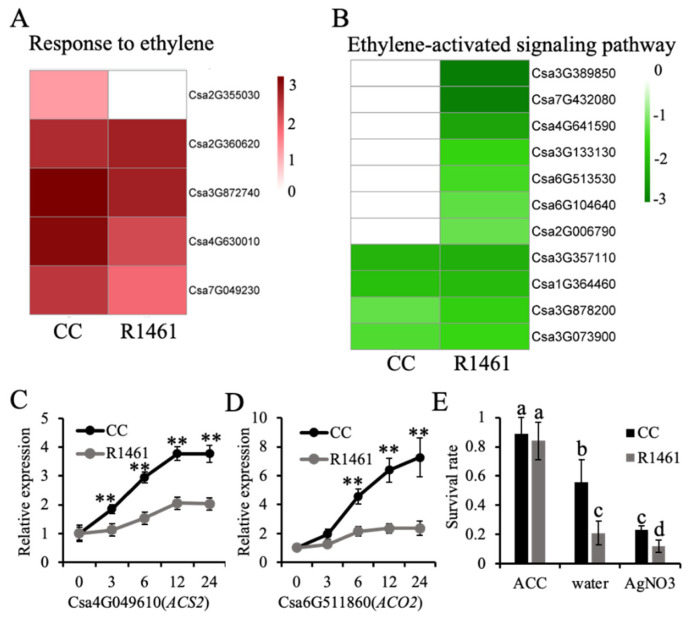
Ethylene confers chilling tolerance in cucumber. (**A**,**B**) Heat maps of the expression levels of genes belonging to “response to ethylene” (**A**) and “ethylene-activated signaling pathway” (**B**). Chilling-induced expression changes are shown, expressed as log_2_FC based on RNA-seq data. (**C**,**D**) Relative expression levels of *ACS2* (**C**) and *ACO2* (**D**) in 16 days old seedlings of CC and R1461 under chilling stress. Expression at 0 h was set to a value of 1. Data are the means ± SD of three biological replicates consisting of 3–6 individual seedlings each. ** means *p* < 0.01, Student’s *t*-test. (**E**) Survival rates of CC and R1461 under 4 days of chilling treatment and 7 days of recovery with 100 µM ACC, water, and 100 µM AgNO3 treatment. Data are the means ± SD of three biological replicates consisting of 30 seedlings each. Different letters above the bars indicate significant differences among samples, as determined by one-way ANOVA/Duncan (*p* < 0.05).

**Figure 4 ijms-23-12834-f004:**
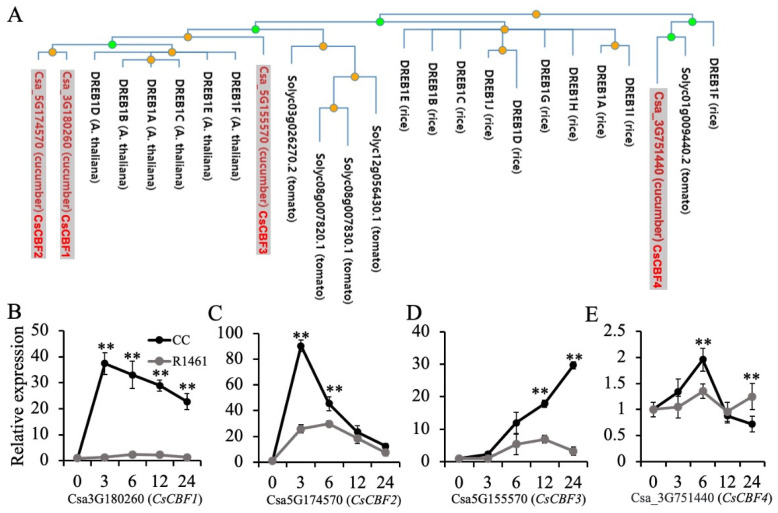
Expression of *DREB1*/*CBF* genes in CC and R1461 under chilling treatment. (**A**) Phylogenetic tree analysis of DREB1/CBF proteins from cucumber, rice, tomato, and *Arabidopsis*. The tree is from PhyloGenes (http://www.phylogenes.org) (accessed on 20 August 2022). using the Arabidopsis *DREB1B* gene as a query on genes from Arabidopsis, tomato, and rice. (**B**–**E**) Relative expression levels of *CsCBF1-4* genes in CC and R1461 under chilling treatment, analyzed by RT-qPCR. Expression at 0 h was set to a value of 1. Data are the means ± SD of three biological replicates consisting of 3–6 individual seedlings each. ** means *p* < 0.01, Student’s *t*-test.

**Figure 5 ijms-23-12834-f005:**
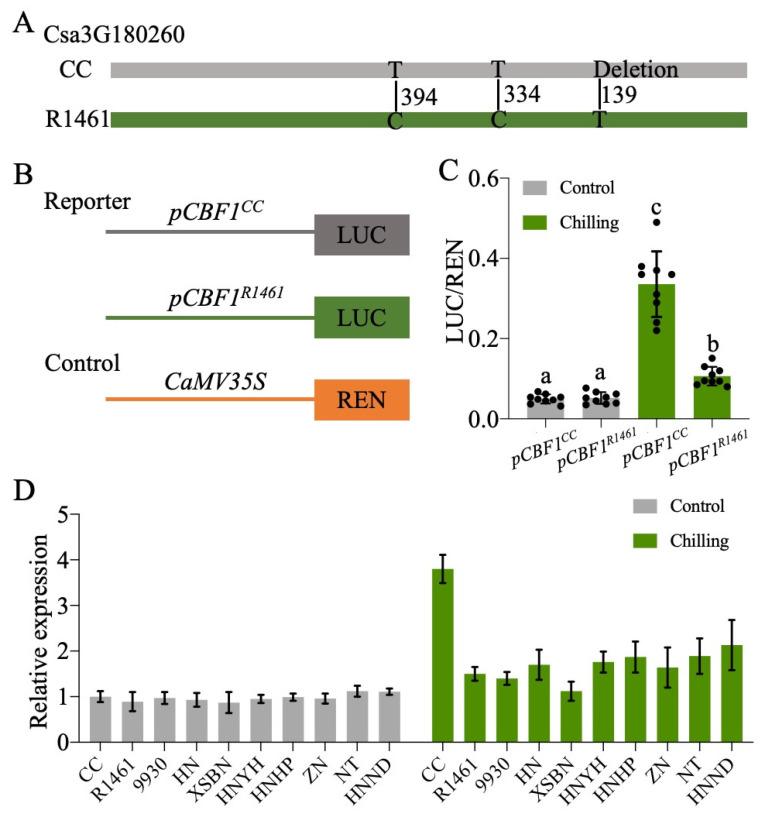
Association of natural polymorphisms in the promoter region of *CBF1* with chilling responses. (**A**) Sequence variations of *CBF1* in CC and R1461. Positions were defined relative to the transcription start site of *CsCBF1*. (**B**) Schematics of the reporter and control constructs used in the dual-luciferase expression assay. LUC, *Firefly* luciferase; REN, *Renilla* luciferase. (**C**) Effects of the polymorphisms on the activity of the *CBF1* promoter. *pCBF1^CC^::LUC* and *pCBF1^R1461^::LUC* were expressed in protoplasts and incubated at 6 °C and 22 °C for 6 h, respectively. Relative reporter activity (LUC/REN) was calculated. The points represent the mean values of 9 independent biological repeats. Different plants growing at different times were used for each biological repeat. Different letters above the bars indicate significant differences among samples as determined by a one-way ANOVA/Duncan (*p* < 0.05). (**D**) RT-qPCR of *CsCBF1* in CC and 9 chilling-susceptible cucumber lines after 0 h (Control) and 3 h (Chilling) of chilling treatment. Expression at 0 h in CC was set to a value of 1. Data are the means ± SD of three biological replicates consisting of 3–6 individual seedlings each.

## Data Availability

Not applicable.

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
