# Peer review of "Transcriptomic Responses to Chilling Reveal Potential Chilling Tolerance Mechanisms in Cucumber"

_ijms, 2022, doi:10.3390/ijms232112834_

Round 1

Reviewer 1 Report

The manuscript should be very carefully revised for grammar, style and proofreading mistakes.

Line 39 The abbreviations of plant hormones should be explained.

Other abbreviations should be arranged according the guide for authors.

Latin names of plants should be written in italics.

Student's test is called by the pseudonym of it's developer, so the the first letter should be capital. 

Reviewer 2 Report

Unfortunately, I recommend the rejection of the manuscript.

 First of all, the manuscript is written in a difficult-to-understand manner. English is poor and it causes that in many places the text is not fully understood.

 It is not known why some experiments were performed on seedlings grown for 11 days (4 days of chilling + 7 days of recovery), some other experiments were executed before chilling and just after 48 hours of chilling, and finally, protoplasts were used in some others experiments. Some seedlings were grown in a mixture of soil and vermiculite, while some others were in ½ Murashige and Skoog medium. Why? Moreover, it is not known how many days seedlings were grown before they were used for chilling treatment.

 The section “Results” is difficult to understand; especially the description of the results of the transcriptomics.

 Generally, there are so many ambiguities in the text that it is impossible to list all of them in this review. In my opinion, first, the manuscript should be written again in good scientific English and in a much easier-to-understand manner.

 Some examples of mistakes

                    The title is difficult to understand. Transcriptomic responses?

                    Abstract: Abbreviations without explanations. Not clear sentences, for example, lines 20-21, what does mean that chilling tolerance disappeared?

                    Keywords should not be the same as words already used in a title. Abbreviations should not be used in the keywords.

                    The Introduction does not give enough background and is difficult to understand, for example, lines 47-54.

                    Chilling, freezing, cold, and low temperatures are freely used in the Introduction. Such different and not precise wording causes confusion.

                    What does mean “miR164-NAM3 module” in line 53 or “hydrogen sulfide module” in line 72?

                    The information given in the first sentence of section 4.1 is completely not important from the scientific point of view. It should be placed only in the Acknowledgment.

Reviewer 3 Report

Dear authors, I have split my suggestions and corrections into two parts, “Content” and “Language” In the cases, where I give for a line only an expression in quotation marks and do not write more to it, this means, the given expression should replace the one that you are using, thereby either correcting a mistake in your text or clarifying something which is otherwise unclear to the reader

Content

l. 16: „the hormones“: please specify which hormones (except ethylene)

l. 18-19: important words missing, and: ACC is the direct precursor of Et, not of Et-biosynthesis. I suggest rephrasing to: „Exogenously applying ACC, the precursor of ethylene, improved …, while exogenous application of the ethylene inhibitor AgNO3 impaired…“

l. 39: “auxins”, “brassinosteroids” (since this a classes of hormones not single hormones). Please also type out the abbreviations for the two hormones for which you do not do it yet; salicylic acid and jasmonic acid (or “jasmonates”, if you want to include other derivatives of jasmonic acid in this summary)

l. 49: “ethylene-insensitive” makes it here confusing for the reader, please leave this out (even though the gene names contain “Ethylene Insensitive X”, due to the properties of respective KO-mutants). I suggest: “loss of function of ethylene signaling genes ETR1, EIN2, … enhanced” or similar.

l. 63: “CsCBF1 and CsCBF2 transcription was rapidly increased…” (I assume, this is what you mean.)

l. 64: “CsCBF3 was less sensitive in transcriptional regulation to…”
l. 72: Your reference [1] here is the wrong reference number. Please correct. Also, I assume what you mean is “the auxin and hydrogen sulfide signaling module”. If so, please insert this word.

l. 80: Which `pathway`?, “signaling pathway” or “biosynthesis”? Please write this more precise.

l.81: Transcriptional regulation of what? Please clarify.

l. 93: Please explain here, why you chose line R1461. (I assume for its medium-to-low survival rate?)

l. 103 and l.105 = Figure 1B and 1D; figure legends: Are these three biological replicates (in l.103 and in l.105) consisting of 30 seedlings each? Please specify here in these figure legends.

l.104: Please write only one asterisk in front of “p < 0.05” and in the figure 1B, because p < 0.05 is the highest possibly usable p-value cut-off, i.e. the lowest acceptable significance level of the t-test.

l. 106: Please write which post -hoc test after the 1-way-ANOVA you have used to be able to determine the different letters above the four bars. The overall ANOVA p-value cannot give you that, it only states, THAT there is a difference between some of the four bars, not specifying which.

Supplement Fig. 1: Please specify in the legend the n of the biological replicates. If a biological replicate contained more than one seedling than specify also this in this figure legend.

l. 115 and l. 117: “Figure S2” (not S1)

l.116: The three replicates of CC 3h chilling (Figure S2) are not really “clustered together”. Please mention this spreading of data in this case.

Figure 2 D: Please also write R1461 and CC on the y-axis of this graph. Please replace “Count” by “Gene count” to clarify this for the reader.

l.152: Do you mean log2FC or log10FC? Please specify.

l. 154: The description of x- and y-axis must be swapped, they are wrongly assigned at the moment.

l. 143: Please write: “the distinct chilling response gene regulation in these two BP terms…” or something similar. (Because you describe the number of genes, not the specific identity of the genes in these two BPs in the two cucumber lines in the text before.)

l.147: Incorrect, since “auxin-activated signaling pathway” is also enriched in CC in Fig 2D.

l. 167: Incorrect. Please correct to: “in three of these four genes is less than in CC”. “Suggesting”:

Since the expression of four of the five genes is also upregulated in R 1461 (in one case even higher than in CC!) the influence of these four genes on the chilling tolerance is in all likelihood small.

l. 171: That the fold-change of down-regulation is less in CC is in your heat map only visible for two genes, not for of all four commonly down-regulated genes. Please correct this sentence, taking this into account.

l. 174 and l. 206 and l. 256: The correct term is “RT-qPCR” (see e.g. the MIQE standards), please write it like this. The quantitative (= “q-”) of these two reactions is not the Reverse Transcription, but the subsequent PCR on the cDNA. RT stands for `Reverse Transcription´.

l.180 ff.: Please write in the text about the water treated control because this is significant for the reader understanding your conclusion here.

l. 191: . How old were the plants used? Please specify. Are the three biological replicates three individual plants (or seedlings) or is each replicate consisting of several plants? Please specify this here and in l. 193. As above: Please write only one asterisk in front of “p < 0.05” and in the figures 3C and 3D, because p < 0.05 is the highest possibly usable p-value cut-off, i.e. the lowest acceptable significance level of the t-test.

l. 194-195: As above, please write which post -hoc test after the 1-way-ANOVA you have used to be able to determine the different letters above the six bars. The overall ANOVA p-value cannot give you that, it only states, THAT there is a difference between some of the six bars, not specifying which.

l. 203-205: Please reference also in l. 204 (regulation of CsCBF1) a table and mark the genes CsCBF1 and CsCBF2 as such in the respective tables, as well as their fold change in regulation, e.g. in color in the tables.

l. 208: Please mark in Figure 4 additionally with these name, which genes are CsCBF1-4

l. 211: You can only write “changed significantly”, when you have performed a statistic, in this case for the time point-values of R1461 along the time course of this line alone, which you do no give. Provide this statistical analysis or omit “significantly”. Additionally, please state then, that also for CsCBF3 there is a change in expression level in line R1461, e.g. at 6 h and 12 h, which is clearly visible in Fig. 4D.

l. 220: As above: Are the three biological replicates three individual plants  (seedlings) or is each biological replicate consisting of several plants? Please write only one asterisk in front of “p < 0.05” and in the figures 3C and 3D, because p < 0.05 is the highest possibly usable p-value cut-off, i.e. the lowest acceptable significance level of the t-test.

l. 228: What are “the other regulators” that you have in mind? Please specify.

Figure 5C and l. 237, referring to it:  the color-coding (green for chilling) is wrong in the bars of the figure: The 3rd and 4th bar from the left would have to be green. 

l. 246: “that the influence on gene expression by the polymorphisms in CsCBF1 was specific to chilling…”

l. 256: As in the cases above, please write which post -hoc test after the 1-way-ANOVA you have used to be able to determine the different letters above the four bars. The overall ANOVA p-value cannot give you that, it only states, THAT there is a difference between some of the four bars, not specifying which.

l. 254: How long was the chilling treatment at 6 °C for Fig. 5C?

l. 255: Please write precisely, what a biological repeat is here in your experiment, which deals with transfection of Arabidopsis mesophyll protoplasts. Meaning: at which level of the experiment was the biological repetition of such a biological repeat?

l. 258: Are the 3 biological replicates 3 individual plants?

l. 266: Given that this is merely  based on changes in RNAseq- and RT-qPCR- results and does not involve subsequent analysis of e.g. KO-lines for respective genes the expression “likely confer” is too strong . I suggest: “…revealed a variation of gene expression that could contribute to differences in chilling tolerance…” or something very similar to this.

l. 267: “285 and 839 DEGs” would be correct.

l. 269: “confer chilling tolerance”: also the genes with down-regulated transcript levels in chilling?

l. 278: “Figure 3C, D”

l. 291: Please omit “involved in ethylene signaling”, because that is only known of the second of the two BP terms, not of both.

l. 294: “ethylene signaling and biosynthesis in these two… possibly contributing to chilling tolerance”

l. 303: Not “Figure 5”, but “Figure 4B-D”

l. 320: How long were the seedlings grown in 22 °C before the respective chilling treatments? Please specify this here.

l. 333: What are the “some modification” to the protocol given in [29]. Please describe the modifications.

l. 340: “replicates including more than 3 seedlings”: Does this mean: “replicates consisting of 3-x seedlings”? Than please write this and give also a number for this x as upper limit.

l. 344 and also l. 352 : Are the respective “3 biological replicates” in these two text lines 3 leaves or does each biological replicate consist of (3?) leaves of 3 individual plants? Please clarify this in both instances.

l.358: What are these “Three independent replicates”: Are they e.g. three technical qPCR-replicates of each of the three biological replicates? Please clarify.

l. 370: “upstream of the CsCBF1 (…) transcription start site…”

l. 394-396: Since the first and the last author share the same first name and the same surname (l. 3) please specify here, which of them did what; e.g. by writing here a specifier in parenthesis “Xiang Wang (first author)” or “Xiang Wang (last author)”.

Language

l. 21: “the difference …between CC and R1461 disappeared”

l. 33: “limiting its global” 

l. 38: “hormones and their respective signaling pathways”

l. 42-43: “induces”, “1-aminocyclopropane-1-caboxylic acid (ACC)”

l. 58: “play a role as a hub”

l. 82: “associated with”

l. 94: “chilling treatment (6 °C)”

l. 96: ”Furthermore, when seedlings of CC…”

l. 97: Please omit “obviously”.

l. 126: “and that these DEGs”; please omit the word “strongly” (no proof of that)

l. 131: “ “Response to chitin” and…,respectively.”

l. 140: “belonging”

l. 142: “plays”

l. 169: “…“…pathway”. Four of the genes herein were…”

l. 198: “Overexpression of”

l. 202: “was close”

l. 213: “to their distinct”

l. 222: “Given the... and that the overexpression…tolerance [25], we then determined…”

l. 233: “Firefly Luciferase (LUC)”

l. 254: Please omit “respectively”, it is wrong here.

l. 265: “…in a chilling tolerant (CC) and a susceptible line (R1462) …”

l. 272: “However, it was not tested in these studies whether …. plays a role…”

l. 274: “was enriched in the chilling tolerant line CC (Figure 2C)…”

l. 275: “was enriched in the chilling susceptible line…”

l. 291: The expression “were response” here is totally unclear, please omit it. Do you maybe mean: “We identified a chilling-responsiveness of transcripts of the two BP terms …”?

l. 313: “contributing to cucumber chilling tolerance”

l. 358: Please use the term “thermo cycler” instead of the colloquial “PCR machine”.

l. 386: “provides more comprehensive information on…”
